# Variational Autoencoding of Dental Point Clouds

**Johan Ziruo Ye**                                    *Johan.Ye@3shape.com*
*Technical University of Denmark*
*3Shape*

**Thomas Ørkild**                                    *Thomas.Orkild@3shape.com*
*3Shape*

**Peter Lempel Søndergaard**                          *Peter.Soendergaard@3shape.com*
*3Shape*

**Søren Hauberg**                                    *sohau@dtu.dk*
*Technical University of Denmark*

**Reviewed on OpenReview:** *https://openreview.net/forum?id=nH416rLxtI*

## Abstract

Digital dentistry has made significant advancements, yet numerous challenges remain. This paper introduces the *FDI 16* dataset, an extensive collection of tooth meshes and point clouds. Additionally, we present a novel approach: *Variational FoldingNet (VF-Net)*, a fully probabilistic variational autoencoder for point clouds. Notably, prior latent variable models for point clouds lack a one-to-one correspondence between input and output points. Instead, they rely on optimizing Chamfer distances, a metric that lacks a normalized distributional counterpart, rendering it unsuitable for probabilistic modeling. We replace the explicit minimization of Chamfer distances with a suitable encoder, increasing computational efficiency while simplifying the probabilistic extension. This allows for straightforward application in various tasks, including mesh generation, shape completion, and representation learning. Empirically, we provide evidence of lower reconstruction error in dental reconstruction and interpolation, showcasing state-of-the-art performance in dental sample generation while identifying valuable latent representations[1].

## 1 Introduction

Recent advancements and widespread adoption of intraoral scanners in dentistry have made micrometer-resolution 3D models readily available. Consequently, the demand for efficiently organizing these noisy scans has grown in parallel. To this end, we propose a variational autoencoder (Kingma & Welling, 2014; Rezende et al., 2014) specifically designed for point clouds, enabling the identification of continuous representations. This approach effectively captures the continuous changes and degradation of teeth over time.

Our solution is a probabilistic latent variable model that ensures a one-to-one correspondence between points in the observed and generated point cloud. This one-to-one connection throughout the network allows for optimization of the original variational autoencoder objective. This is achieved by projecting the point cloud onto an intrinsic 2D surface representation, which allows for efficient sampling and also discourages storage information about the overall shape within this space. These 2D projections impart a strong inductive bias, proving highly beneficial when the input point cloud and the 2D surface share topology. Notably, this also bottlenecks the model, preventing it from learning the identity mapping. Specifically, *Variational Foldingnet (VF-Net)* learns a projection from the 3D point cloud input down to 2D space, which then is deformed back to reconstruct the input point cloud. Finally, these projections facilitate mesh generation without further

---

[1]Code available here

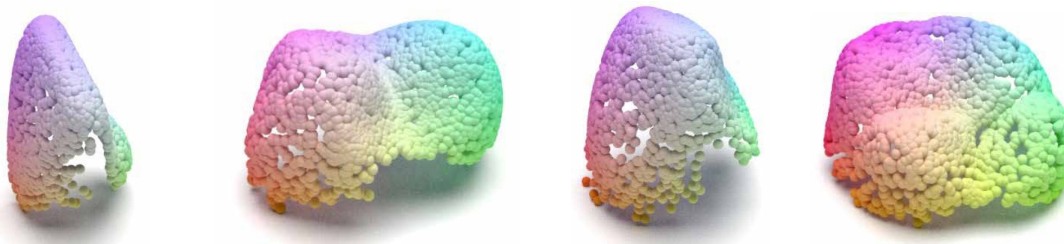

Figure 1: VF-Net teeth samples, generated by our probabilistic variational autoencoder for point clouds. Note the wide variety in the samples which retain anatomical details in its cusps/fissure composition.

training, as well as straightforward shape completion and shape extrapolation, all without compromising the quality of the learned representations (see Fig. 1 for samples). Previous point cloud models generally lack one-to-one correspondence throughout the network due to their invariant architecture design. Instead, they evaluate reconstruction error using *Chamfer distances* (CD) (Barrow et al., 1977) defined as

$$\textsc{Chamf-Dist}(\mathbf{x}, \mathbf{y}) = \frac{1}{|\mathbf{x}|} \sum_{i=1}^{m} \min_{y_j \in \mathbf{y}} \|x_i - y_j\|_2 + \frac{1}{|\mathbf{y}|} \sum_{j=1}^{n} \min_{x_i \in \mathbf{x}} \|y_j - x_i\|_2, \tag{1}$$

where $m$ and $n$ are the number of elements of $\mathbf{x}$ and $\mathbf{y}$ respectively. This metric solves the invariance problem. However, it also poses a new one: *The Chamfer distance does not readily lead to a likelihood, preventing its use in probabilistic modeling.* For instance, when used in the Gaussian distribution, the function $\mathbf{x} \mapsto {}^1/c \exp(-\textsc{Chamf-Dist}^2(\mathbf{x}, \mu))$ cannot be normalized to have unit integral due to the explicit minimization in Eq. 1. Consequently, previous latent variable models are closer to regularized autoencoders than the variational autoencoder. Since our model ensures one-to-one correspondence between points in the point clouds, we can easily build a proper probabilistic model.

Moreover, to encourage further research, we release a new dataset, the *FDI 16 Tooth Dataset*, providing a large collection of dental scans, available as both meshes and point clouds[2]. This dataset provides real-world representations with planar topology. We consider this an excellent compromise between high-quality computer-aided design (CAD) models and sparse LiDAR scans (Chang et al., 2015; 2017; Caesar et al., 2020; Armeni et al., 2016). In digital dentistry, significant challenges are found in diagnostics, tooth (crown) generation, shape completion of obstructed areas of the teeth, and sorting point clouds, etc.

**In summary**, we present the first fully probabilistic variational autoencoder for point clouds, VF-Net, characterized by a highly expressive decoder with state-of-the-art generative capabilities. All while learning compressed representations and being adaptable for shape completion tasks. Furthermore, we release a dataset of 7,732 tooth meshes to facilitate further research on real-world 3D data.

## 2 Related work

We focus on point cloud representations of 3D objects, but there are many alternative methods of representation including voxel grids (Zheng et al., 2021; Wu et al., 2018), multi-angle inference (Wen et al., 2019; Han et al., 2019), and meshes (Alldieck et al., 2019; Wang et al., 2018; Groueix et al., 2018). A major paradigm in neural networks for point clouds is to remain permutation and cardinality invariant. In terms of encoder-decoder models, this frequently leads to designs without a one-to-one correspondence between inputs and outputs (Yang et al., 2018; Groueix et al., 2018). This becomes an obstacle in adapting the variational autoencoder to point clouds. Accordingly, other methods have become prominent, including GANs (Li et al., 2018; 2019), diffusion models (Zhou et al., 2021; Zeng et al., 2022; Zhou et al., 2023), and traditional autoencoders (Achlioptas et al., 2018; Groueix et al., 2018; Pang et al., 2021).

**Existing Point Cloud Variational Autoencoders.** Previous attempts to design a variational autoencoder for point clouds frequently relies on Chamfer distances as an approximation of the reconstruction

---

[2]Data available here

term in the standard evidence lower bound. Consequently, these VAEs fail to evaluate a likelihood, a key characteristic of VAEs. This includes works like EditVAE, which aims to disentangle each point cloud into smaller parts. For each disentangled part, they use the Chamfer distance individually and a superquadric loss that consists of another Chamfer distance term and a regularization term to prevent overlapping parts (Li et al., 2022). The Venatus Geometric Variational Auto-Encoder (VG-VAE) introduces a Geometric Proximity Correlator module to better capture local geometric signatures. However, their work also relies on the Chamfer distance as the reconstruction term. Another latent variable model for point clouds is SetVAE (Kim et al., 2021), which uses transformers to process point clouds as sets. Their primary novelty being the introduction of a latent space with an enforced prior inside the transformer block. These transformer blocks are then stacked to form a hierarchical variational autoencoder (Sønderby et al., 2016), which complicates evaluation of its representations. However, the SetVAE also approximates their reconstruction loss via Chamfer distances. Without explicit likelihood evaluation, these models become closer to a regularized autoencoder than the variational autoencoder.

**Other Generative Models.** On the other hand, LION (Zeng et al., 2022) is a latent diffusion model (Rombach et al., 2022) that maintains a one-to-one mapping throughout the network, allowing for probabilistic evaluation. However, they only implicitly utilize this by optimizing an L1-loss. Similar to our work, they encode their points in a separate space, but instead of bottlenecking this, they map them to a higher dimensional space. This, unfortunately, leads to information about the shape being stored here, preventing direct sampling/modification to the embedded points in this space. Similarly to SetVAE, evaluating the quality of representations in LION, a hierarchical latent variable model, poses challenges. Recently, Zhou et al. (2023) presented FrePolad, another latent diffusion model. Their primary novelty is the introduction of the frequency rectification module that better captures high-frequency signals in point clouds. They train their model via a modified VAE loss to account for frequency rectified distances. One fully probabilistic work is Point-Flow (Yang et al., 2019). PointFlow utilizes a continuous

| | Generative | Mesh | Completion | Probabilistic | Representations |
|---|---|---|---|---|---|
| SetVAE | ✓ | ✗ | ✓ | ✗ | ✗ |
| LION | ✓ | ✗ | ✗ | ✗ | ✗ |
| FrePolad | ✓ | ✗ | ✗ | ✗ | ✓ |
| PointFlow | ✓ | ✗ | ✓ | ✓ | ✗ |
| DPM | ✓ | ✗ | ✓ | ✓ | ✗ |
| PVD | ✓ | ✗ | ✓ | ✓ | ✗ |
| FoldingNet | ✗ | ✓ | ✓ | ✗ | ✓ |
| VF-Net (ours) | ✓ | ✓ | ✓ | ✓ | ✓ |

Table 1: VF-Net is a generative model (Generative) for point clouds, but it can generate meshes without additional training (Mesh) and do simple shape completion (Completion). It is also fully probabilistic (Probabilistic) and can identify interpretable lower-dimensional representations (Representations).

normalizing flow (CNF) both as a prior and decoder, similar to approaches previously applied to images (Kingma et al., 2017; Sadeghi et al., 2019). Intuitively, one CNF models the distribution of shapes, while the other models the point distribution given the shape. In a comparable way, VF-Net's encoder maps to a global latent space, with point encoding projections providing a latent mapping for each input point. However, PointFlow's two CNFs are trained separately, whereas VF-Net trains them simultaneously, resulting in a more integrated and efficient process. PointFlow is unfortunately very slow to train (Kim et al., 2021). On our full proprietary dataset, PointFlow would have required 200 GPU days of training. Thus, we excluded it from our baselines. Diffusion models such as diffusion probabilistic model (DPM) (Luo & Hu, 2021) and point-voxel diffusion (PVD) (Zhou et al., 2021) present two diffusion models for the point clouds, especially PVD generates accurate new samples. However, diffusion models do not find compressed structured representations of the data as our VF-Net does; see table. 1 for a model property overview.

**Digital Dentistry.** In computational dentistry, extrapolating the tooth's obstructed sides is a well-known task. Qiu et al. (2013) presents an attempt to use classic computational geometry methods. They attempt to reconstruct the missing parts of the distal and mesial sides of the tooth. This leads to a very smooth extrapolation, which performs well. Several works within dentistry take this a step further, e.g., attempting to extrapolate not just the sides but also the roots of the teeth (Wei et al., 2015; Zhou et al., 2018; Wu et al., 2016). We are optimistic that our model could adapt to such a task given that dental cone beat computed tomography (CBCT) of the dental roots was available in the training data. Unfortunately, CBCT scans are expensive and rare; thus, we do not have a large enough dataset for neural network training.

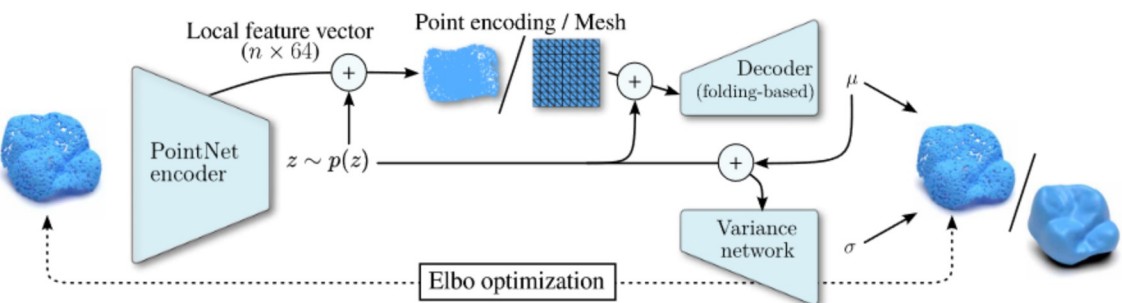

Figure 2: VF-Net is a variational autoencoder with a normalizing flow prior over the shape latent. Individual points are projected to 2D space, establishing a one-to-one connection and facilitating mesh generation and shape completion. The decoder follows FoldingNet's with added residual connections, while the variance network consists of 3 folding modules as introduced in FoldingNet.

## 3 Variational Point Cloud Inference

**Background: FoldingNet.** To handle varying sizes and arbitrary order in point clouds, a common strategy is to employ neural networks exhibiting invariance to changes in cardinality and permutation, as proposed by Qi et al. (2017) in PointNet. FoldingNet employs a very similar encoder, $e$, that operates independently on each point of the point cloud to identify a latent code, $\mathbf{z}$. Subsequently, the folding-based decoder, $f : \mathcal{Z} \times \mathbb{R}^2 \to \mathbb{R}^3$, "folds" a chosen constant base shape with points, $\mathbf{c}$, according to the latent code. In our case, the base shape is a constant uniform grid in the two-dimensional planar patch $[-1, 1]^2$ (Yang et al., 2018). Both the encoder, $e$, and the decoder, $f$, are jointly trained to minimize the reconstruction error approximated via Chamfer distances (1),

$$\mathcal{E} = \text{CHAMF-DIST}\left(\mathbf{x}, f(e(\mathbf{x}), \mathbf{c})\right). \tag{2}$$

This ensures invariance to cardinality and permutation changes, although it complicates variational inference extensions. A variational autoencoder yields a distribution for each input point (Kingma & Welling, 2014; Rezende et al., 2014). However, FoldingNet and most current permutation-invariant neural networks do not have a correspondent output for each individual input point in a point cloud.

### 3.1 The Variational FoldingNet

Motivated by unsupervised probabilistic representation learning's benefits across many tasks, including *generative modeling* (Kingma & Welling, 2014; Rezende et al., 2014; Dinh et al., 2017; Ho et al., 2020), *out-of-distribution detection* (Nalisnick et al., 2019; Havtorn et al., 2021), *handling missing data* (Mattei & Frellsen, 2019) etc, we introduce Variational FoldingNet (VF-Net). Architecturally, VF-Net closely resembles FoldingNet, employing a PointNet encoder, with the decoder structure mirroring that of FoldingNet. For a complete overview, consult Fig. 2.

The major technical innovation is the introduction of a novel projection for each input point into the planar space, defined as $\mathcal{G} = [-1, 1]^2$. Let $\mathbf{x}$ be a point cloud of points $x_i, \ldots, x_n$. Each projection corresponds to each point $g_i, \ldots, g_n$ in the set $\mathbf{g}$. We will refer to these projections as our point encodings. It is important to note that the point encodings are not constrained by any prior distribution. Decoding these point encodings instead of a static planar patch establishes a one-to-one correspondence throughout the entire network, a necessity for evaluating likelihoods using the classical variational autoencoder objective. As VF-Net learns the point projections from $\mathbf{x}$, the projected points, $\mathbf{g}$, are now dependent on $\mathbf{x}$. The folding of the point encodings, $f(\mathbf{z}, \mathbf{g})$, continues to be governed by the latent parameter vector $\mathbf{z}$ predicted by the PointNet encoder, $e$. The optimal projections are thus given by

$$g_i = \underset{g' \in \mathcal{G}}{\arg\min} \|x_i - f(\mathbf{z}, g')\|^2, \tag{3}$$

where $g_i \in \mathbf{g}$. We use a neural network to amortize the calculation of $\mathbf{g}$ such that the encoder network outputs both $\mathbf{g}$ and the distribution of $\mathbf{z}$. By enabling the model to adjust the point encoding, we circumvent

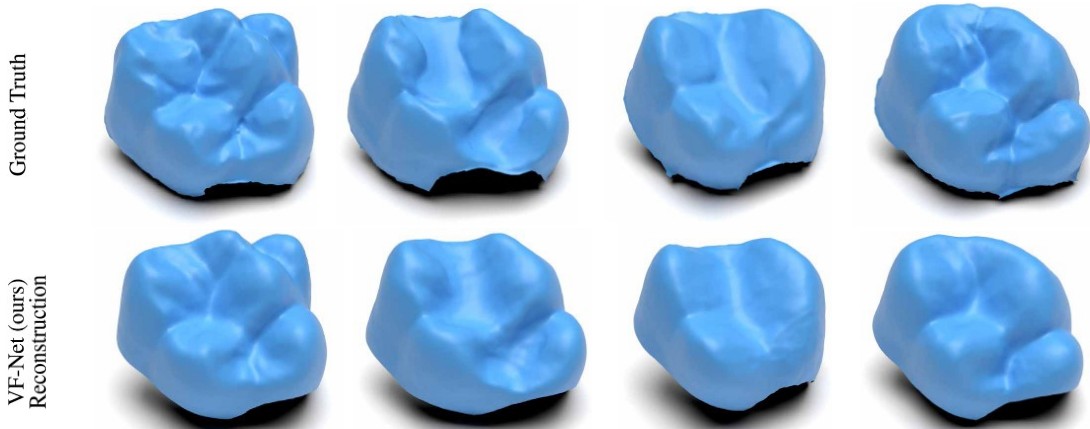

Figure 3: *Top:* Mesh data samples from our released FDI 16 dataset and their corresponding VF-Net reconstructions. Note the large variety in health conditions between the teeth.

the need for optimizing through costly Chamfer distances. Furthermore, the learned projections allow the point encodings to adapt to their input, mitigating common pitfalls observed in FoldingNet, see Fig. 4.

With a one-to-one point correspondence established across the network, we optimize our model using traditional variational autoencoder methods. In this context, the variational extension aligns closely with traditional methods, yet with a notable adjustment, the evaluation of likelihood now also depends on the projected points $p(\mathbf{x}) = \int p(\mathbf{x}|\mathbf{z}, \mathbf{g}) \, p(\mathbf{z}) \, \mathrm{d}\mathbf{z}$. This integral remains intractable, and approximations are necessary. Following conventional variational inference (Kingma & Welling, 2014; Rezende et al., 2014), an evidence lower bound (Elbo) on $p(\mathbf{x})$ is given by

$$\mathcal{L}(\mathbf{x}) = \mathbb{E}_{q(\mathbf{z}|\mathbf{x})}[\log p(\mathbf{x}|\mathbf{z}, \mathbf{g})] - \mathrm{KL}(q(\mathbf{z}|\mathbf{x})\|p(\mathbf{z})), \tag{4}$$

where $q(\mathbf{z}|\mathbf{x})$ is an approximation to the posterior $p(\mathbf{z}|\mathbf{x})$, which is assumed to follow a gaussian distribution. Note that Eq. 3 is implicitly optimized in the likelihood term of the ELBo. Most current point cloud models replace the likelihood with a Chamfer distance, making the models closer to regularized autoencoders (Yang et al., 2018; Groueix et al., 2018; Kim et al., 2021). This design loses one-to-one correspondences between input and output, making likelihood evaluation difficult. In particular, no suitable normalization constant can be derived for probabilistic distributions using Chamfer distances.

Our novel method for probabilistic evaluation for 3D reconstruction networks avoids the computationally expensive Chamfer distance (1). In supplementary Fig. S1, we empirically demonstrate that our projections can effectively replace Chamfer distances. We observe that the two metrics closely align, with Euclidean distances acting as an upper bound that tightens with improved reconstruction precision.

During the evaluation of the Elbo loss, we use a multivariate student-t distribution with isotropic variance and three degrees of freedom as the reconstruction term. This choice helps to decrease emphasis on outliers and instead focus more on the majority of the data points (Takahashi et al., 2018). $p(\mathbf{x}|\mathbf{z}, \mathbf{g}) = \mathrm{Student\text{-}t}(\mathbf{x}|f(\mathbf{z}, \mathbf{g}), \sigma^2(\mathbf{z}, \mathbf{g})\mathbf{I}, \nu)$, where $f : \mathcal{Z} \times \mathbb{R}^2 \to \mathbb{R}^3$ and $\sigma^2 : \mathcal{Z} \times \mathbb{R}^2 \to \mathbb{R}_+$ are neural networks. No major changes were made to the generative process. We let a normalizing flow model the prior, $p(\mathbf{z})$, which describes the *shape* of an object (Kingma et al., 2017). Note that this is trained subsequently and no downscaling of the Kullback-leibler divergence term occurs. When the input, $\mathbf{X}$, and the projections, $\mathcal{G}$, share topology, the bias allows for uniform sampling in the planar patch $[-1, 1]^2$. As in FoldingNet, this grid is subsequently deformed according to $\mathbf{z}$. New samples can thus be generated by first sampling $\mathbf{z}$ and then mapping the uniformly sampled grid points through $f$ and $\sigma$,

$$\mathbf{x} = f(\mathbf{z}, \mathbf{g}) + \sigma(\mathbf{z}, \mathbf{g}) \cdot \mathbf{t}, \quad \mathbf{t} \sim \mathrm{Student\text{-}t}(\nu). \tag{5}$$

This also enables straightforward mesh generation as deformations are smooth - points projected closely to each other correspond to points close in output space. Consequently, we can generate meshes by simply defining the facets in the 2D planar space.

## 4  The FDI 16 Tooth Dataset

To improve the state-of-the-art modeling of dental scans, we will release an extensive new dataset alongside this paper under the CC BY-NC-SA 4.0 license. The FDI 16 dataset is a collection of 7,732 irregular triangle meshes of the right-side first maxillary molar tooth formally denoted as 'FDI 16' following ISO 3950 notation (see Fig. 3). These meshes were acquired from fully anonymized intraoral scans primarily scanned using 3Shape's TRIOS 3 scanners. Each tooth in the FDI 16 Tooth dataset was algorithmically segmented from an upper jaw scan by 3Shape's Ortho Systems 2023. As the teeth are a subsection of a full intraoral jaw scan, there will be areas obstructed by the adjacent teeth. The teeth, therefore, constitute open meshes and have clear boundaries with no representation of interior object volume. All tooth meshes are from patients undergoing aligner treatment, and accordingly, aligner attachments will be present in a substantial number of scans. This introduces a bias towards younger individuals, who generally have fewer restorations and dental problems. The top row of Fig. 3 shows examples of such meshes. All scans have been made publicly available fully anonymously as meshes and point clouds at millimeter scale. The teeth have been algorithmically rotated to ensure that the $x$-axis is turned towards the neighboring tooth (FDI 17) while the $y$-axis points in the occlusal direction (direction of the biting surface). Finally, the $z$-axis is given by the cross-product to ensure a right-hand coordinate system.

Dental scans have a diverse set of research applications. This study explores reconstruction, generation of new teeth, representation learning, and shape completion. All of which have different but critical applications in digital dentistry. We believe that the FDI 16 dataset addresses a crucial niche within 3D datasets by offering a dataset that strikes a balance between the highly detailed but idealized CAD scans (Chang et al., 2015) and sparser real-world LIDAR scans (Chang et al., 2017; Caesar et al., 2020; Armeni et al., 2016). Note that any method considered for deployment must be capable of running efficiently on edge devices without a significant performance overhead. This is particularly important as intraoral scanners must function seamlessly even in areas with limited network connectivity.

## 5  Experimental results

We next evaluate VF-Net's performance on point cloud generation, auto-encoding, shape completion, and unsupervised representation learning. Note that FrePolad (Zhou et al., 2023), EditVAE (Li et al., 2022), and VG-VAE (Anvekar et al., 2022) has been excluded from comparison as no public implementation is available.

**Point cloud generation.** To compare sampling performances, we deploy three established metrics for 3D generative model evaluation (Yang et al., 2019). Namely, minimum matching distance (MMD) is a metric that measures the average distance to its nearest neighbor point cloud. Coverage (COV) measures the fraction of point clouds in the ground truth test set that is considered the nearest test sample neighbor for a generated sample. 1-nearest neighbor accuracy (1-NNA) uses a 1-NN classifier to classify whether a sample is generated or from the ground truth dataset, 50%, meaning generated samples are indistinguishable from the test set. Data handling and training details for FDI 16 experiments can be found in supplementary section S1.3 and S1.4, respectively.

Table 2: Across five seeds, VF-Net produces close to as large a variety of teeth as PVD and LION while generating samples much closer to real teeth. MMD has been multiplied by 100.

| Method | MMD($\downarrow$) | | COV(%$\uparrow$) | | 1-NNA(%$\downarrow$) | |
|---|---|---|---|---|---|---|
| | CD | EMD | CD | EMD | CD | EMD |
| Train subsampled | $21.00_{\pm 0.09}$ | $51.53_{\pm 0.06}$ | $49.00_{\pm 0.64}$ | $46.95_{\pm 2.79}$ | $49.83_{\pm 0.68}$ | $50.97_{\pm 0.82}$ |
| SetVAE | $39.00_{\pm 0.78}$ | $66.66_{\pm 0.38}$ | $10.66_{\pm 0.66}$ | $9.52_{\pm 0.27}$ | $97.99_{\pm 0.32}$ | $97.95_{\pm 0.34}$ |
| DPM | $20.71_{\pm 0.10}$ | $51.94_{\pm 0.09}$ | $36.94_{\pm 0.65}$ | $33.28_{\pm 0.65}$ | $70.30_{\pm 0.82}$ | $75.75_{\pm 0.99}$ |
| PVD | $21.58_{\pm 0.03}$ | $51.64_{\pm 0.08}$ | $44.11_{\pm 0.76}$ | $43.23_{\pm 0.92}$ | $62.85_{\pm 0.78}$ | $60.70_{\pm 1.06}$ |
| LION | $22.12_{\pm 0.15}$ | $52.75_{\pm 0.12}$ | $\mathbf{45.12}_{\pm 0.60}$ | $\mathbf{43.32}_{\pm 1.28}$ | $68.56_{\pm 0.73}$ | $66.76_{\pm 0.94}$ |
| VF-Net (Ours) | $\mathbf{20.38}_{\pm 0.09}$ | $\mathbf{49.72}_{\pm 0.04}$ | $42.85_{\pm 0.64}$ | $40.20_{\pm 0.71}$ | $\mathbf{56.31}_{\pm 0.39}$ | $\mathbf{56.05}_{\pm 0.32}$ |

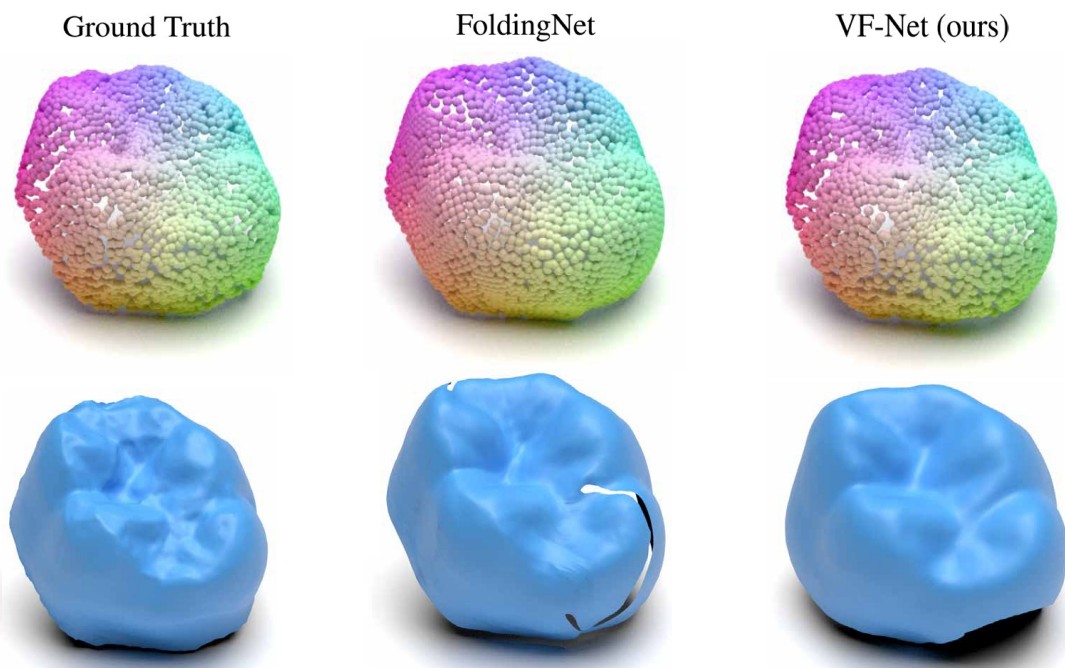

Figure 4: FoldingNet's mesh reconstructions have gaps and highly distorted facets. Conversely, VF-Net's mesh facets are even more regular than the input point cloud, and points in the reconstruction are placed closely resembling its input.

Sampling from VF-Net can be done by sampling a uniform grid in the latent point encodings space, akin to FoldingNet. However, the corners of the uniform grid cause edge artifacts in the generated samples, evident in generated meshes in Fig. S2. This can also be observed in the generated meshes in Fig. 3 and Fig. 4, although it is more difficult to spot. The sampling metrics heavily punish such artifacts. Instead, we trained a minor network similar to the decoder of FoldingNet to predict the point encodings from the latent representation. We emphasize that this is entirely unnecessary for regular sampling. The sampling evaluations across five different seeds can be found in Table 2. The results demonstrate that VF-Net generates much more accurate samples, as evidenced by the significantly lower MMD and 1-NNA scores while being close in diversity to PVD and LION (Zhou et al., 2021; Zeng et al., 2022). Furthermore, sampling is much faster than PVD and LION as VF-Net does not depend on an iterative diffusion process. Note that while MMD is very stable across seeds, the COV and 1-NNA scores may vary.

Outside of the FDI 16 dataset, we also train VF-Net on a proprietary dataset, which includes the remaining teeth from the FDI 16 jaws; see supplementary section S1.5 for training details. However, we did not quantify sampling performance, as sampling evaluation on 40k test samples would be exceedingly computationally expensive. We observe that VF-Net can sample from all major teeth types, incisors, canines, premolars, and molars, see Fig. 1. Additional mesh samples may be found in supplementary Fig. S2.

Table 3: Reconstruction error measured in Chamfer distances (CD) and earth mover's distances (EMD). Note both values have been multiplied by 100.

| Method | FDI 16 Tooth | | All FDIs | |
|---|---|---|---|---|
| | **CD** | **EMD** | **CD** | **EMD** |
| DPM | 10.04 | 43.98 | 5.67 | 35.8 |
| SetVAE | 21.50 | 59.24 | 9.98 | 51.48 |
| LION | 5.35 | 22.85 | 3.02 | 9.66 |
| FoldingNet | 5.26 | 33.67 | 3.43 | 31.25 |
| VF-Net (ours) | **1.21** | **6.30** | **0.97** | **5.30** |

**Point cloud auto-encoding.** We evaluate VF-Net's reconstruction quality to the previously mentioned generative models and FoldingNet. This evaluation was performed on both on FDI 16 dataset and the larger proprietary dataset. Please consult supplementary sections S1.3 and S1.5 for data handling and training details. We compared the reconstruction errors using Chamfer distance and earth mover's distance (Rubner et al., 2000),

$$\text{EMD}(\mathbf{x}, \mathbf{y}) = \min_{\phi:\mathbf{x}\to\mathbf{y}} \sum_{x_i \in \mathbf{x}} \|x_i - \phi(x_i)\|_2. \tag{6}$$

The earth mover's distance measures the least expensive one-to-one transportation between two distributions. However, this is computationally expensive and thus rarely used for model optimization (Wu et al., 2021). The reconstruction errors are presented in Table 3. Point-Voxel Diffusion (PVD) (Zhou et al., 2021) was excluded from comparison due to not returning the same tooth upon reconstruction.

VF-Net achieves a significantly lower reconstruction error than our comparison methods on both the FDI 16 dataset and the proprietary dataset comprising 119,496 teeth, encompassing 32 distinct teeth. As shown in Fig. 4, VF-Net's one-to-one correspondence is evident in its reconstruction. The point placements mimic those in the input point cloud, while FoldingNet's points are evenly distributed. VF-Net and FoldingNet can both generate meshes without any additional training of the model.

However, FoldingNet folds the edge across the tooth to accommodate teeth of different sizes. Besides mesh gaps, this also leads to highly irregular facets that intersect one another. On the other hand, VF-Net can adjust the point encoding area to avoid such artifacts. However, VF-Net's reconstructions often exhibit excessive smoothness and lack the desired level of detail. A common observation in variational autoencoders (Kingma & Welling, 2014; Vahdat & Kautz, 2021; Tolstikhin et al., 2019).

**Variance estimation for point clouds.** Predicted variances from the variance network are shown in Fig. 5, where red indicates a higher variance and green indicates a lower variance within each point cloud. Note that all variances shown are relative intra-point cloud variances. Notably, the network assigns higher variance to the fifth cusp and aligner attachments, features only present in a subset of samples. Furthermore, the border of the mesh tends to be assigned higher variance, likely due to a combination of data loading and segmentation artifacts. When the network is not in doubt about the previously mentioned two factors, the network assigns the highest variance to the occlusal surface. All of which aligns with expectations of areas of the teeth that have the most variance.

**Simulated shape completion.** One significant benefit of the inductive bias from the point encodings is straightforward shape completion and shape extrapolation. In computational dentistry, inferring the obstructed sides of a tooth and reconstructing the tooth surface beneath obstructions such as braces pose a key challenge. Paired

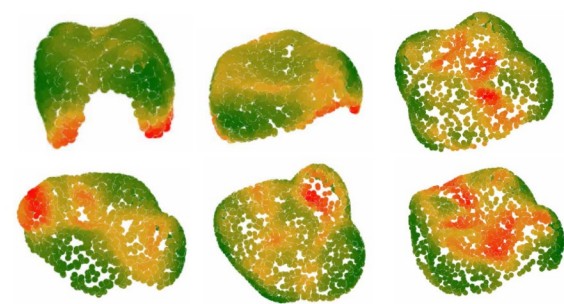

Figure 5: Intra-point cloud relative predicted variance (red is high, green is low). Notably, the carabelli cusp and aligner attachment areas exhibit high variance, two features only present in a subset of individuals.

Reconstructions and the Corresponding Latent Point Encoding

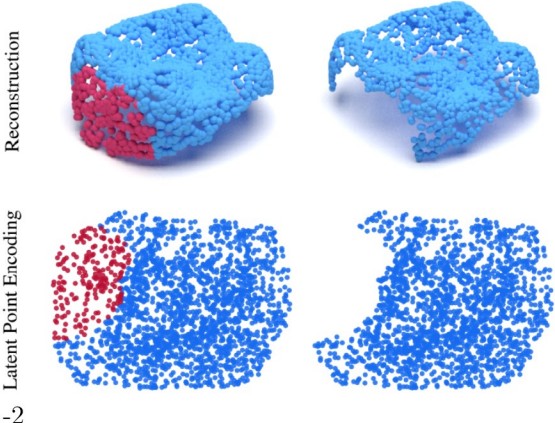

Figure 6: *Left*: Red points are removed from the point cloud. *Right*: Reconstruction and projected point encodings remain highly similar despite point deletion. Sampling the missing area is facilitated by sampling within the corresponding empty region of the latent point encoding.

data of obstructed and unobstructed surfaces is exceedingly rare. Therefore, developing a model capable of extrapolating such surfaces without explicit training is highly desirable. To this end, we simulate the

task by evaluating the interpolation performance of each model. This is done by sampling a point on the outward side of the tooth and deleting its nearest neighbors to a total of 200 points. Selecting a mid-buccal point simulates bracket removal prediction ("Bracket sim") while opting for a lower buccal point simulates the obstructed side prediction ("Gap sim").

An example of a synthetic hole is depicted in Fig. 6, where the red points are to be removed. Both reconstructions and latent point encodings remain highly similar despite the removal of the red points. Extrapolation/interpolation can be performed by sampling in the point encoding space. To quantify the interpolation performance, we calculate the distance from the deleted points to their nearest neighbor in the completed point cloud; see supplementary Sec. S1.7 for more experiment details. To contextualize the performance, we trained several shape completion methods (PVD (Zhou et al., 2021), PoinTr (Yu et al., 2021), VRCNet (Pan et al., 2021)). Since these methods only predict the missing area, a completely fair comparison cannot be made. The results can be found in Table 4, under "Bracket sim" and "Gap sim," simulating the removed bracket

Table 4: Unsupervised generative models in the top half are untrained interpolation, while the bottom half are trained models. All Chamfer distances have been multiplied by 100.

|  | Method | Bracket sim | Gap sim |
|---|---|---|---|
| Unsupervised | DPM | 15.88 | 38.00 |
|  | SetVAE | 11.50 | 13.35 |
|  | FoldingNet | 16.42 | 20.14 |
|  | VF-Net (ours) | **4.35** | **3.55** |
| Supervised | PVD | 2.23 | 2.37 |
|  | PoinTr | **1.84** | **1.83** |
|  | VRCNet | 2.42 | 2.04 |

and the gap between teeth, respectively. Here, VF-Net outperforms its peers when it comes to untrained interpolation, and as expected there is a gap in performance between the trained and untrained methods. Shape completion using LION's latent points from the original tooth contains information about the shape, rendering a fair comparison infeasible.

Table 5: Percentage teeth which had classification prediction increase according to expectation when moved in the tooth wear direction. L, M, H denotes light, medium, and heavy wear respectively.

| Method | L → H | L → M | M → L | M → H | H → M | H → L |
|---|---|---|---|---|---|---|
| FoldingNet | 91.77 | 91.77 | 95.02 | 94.89 | 97.80 | 97.80 |
| VF-Net (ours) | **92.11** | **99.31** | **97.04** | **96.37** | **98.24** | **99.12** |

**Representation learning.** We compare our latent representation to FoldingNet's, as it is the comparison model with the most interpretable latent variables. First, we follow FoldingNet's proposed evaluation method of classifying the input point cloud from the latent space. Using a linear support vector machine (SVM) to classify which tooth from the larger proprietary dataset is embedded, a 32-class problem. Here, the SVM achieves 96.80% accuracy on VF-Net's latent codes compared to 96.36% of FoldingNet. Indicating all global point cloud information is stored in the latent variables, meaning the latent point encodings exclusively contain information about specific points. No information pertaining to the overall point cloud shape is stored in the point encodings. For qualitative assessment, an interpolation between two FDI 16 teeth and an interpolation example between an incisor and a premolar can be found in Fig. 7. Both interpolations exhibit a seamless transition in the latent space; for a more detailed view, see supplementary Fig. S3.

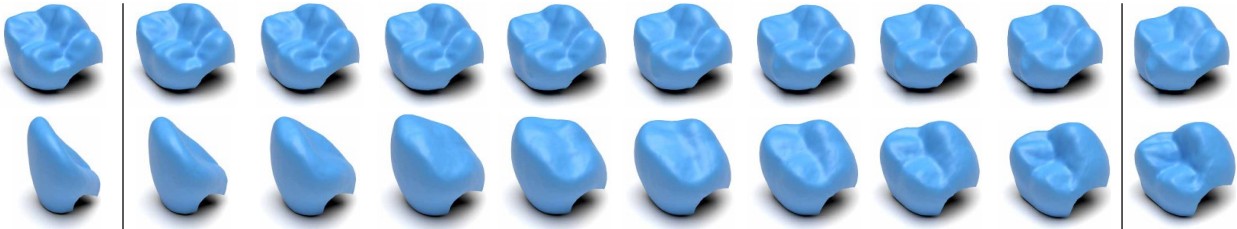

Figure 7: Interpolating between two teeth by interpolating their latent codes using the same mesh decoding.

Removed Tooth Wear           Medium Wear Reconstruction           Added Tooth Wear

Figure 8: Moving in the tooth wear direction in latent space. *Left*: Red areas have higher values than the original. *Middle*: The original reconstruction. *Right*: Blue areas are lower than the original. As the level of tooth wear increases, we observe a gradual smoothing in the occlusal surface.

Next, we attempt to add and remove toothwear; see Fig. 8. We navigate the latent space of VF-Net in the direction of tooth wear or away from it. The direction was determined by calculating the average change in latent representations when encoding 10 teeth from their counterparts with synthetically induced tooth wear. These teeth were manually sculpted to simulate tooth wear; see supplementary Fig. S4. We observe behavior that closely aligns with our expectations of how the tooth would change when adding or subtracting tooth wear.

To quantify the performance, we train a small PointNet model (Qi et al., 2017) on a proprietary dataset of 1400 teeth annotated with light/medium/heavy tooth wear. Subsequently, validate whether a change in the latent space yielded the expected change in classifier prediction. In Table 5, each class denotes the base class before adding/removing tooth wear. For light and heavy, we added and removed tooth wear, respectively, while medium tooth wear teeth were evaluated both when adding/removing wear. The findings presented in Table 5 indicate that VF-Net's latent representations show greater robustness.

**Limitations**. Similar to variational autoencoders in other domains, VF-Net tends to produce overly smooth samples. This characteristic could impact applications such as crown generation, where precise replication of the biting surface is crucial to prevent patient discomfort. Moreover, the model's tendency towards smoothness suggests potential challenges in capturing finer details of teeth, which are essential for comprehensive representation learning.

Until now, the inductive bias from folding a 2D plane to a point cloud has proven highly beneficial. This is only the case when the input point cloud shares topology with the 2D plane. Unfortunately, this inductive bias is not as beneficial when the two topologies differ. We trained VF-Net on ShapeNet data (Chang et al., 2015). The drawback is not evident through the reconstructions; see supplementary Table S1. VF-Net has a low reconstruction error, but LION boasts the lowest. Issues arise when attempting to generate new samples. Due to information of the shape being stored in the latent point encodings, as depicted in

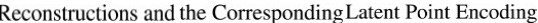

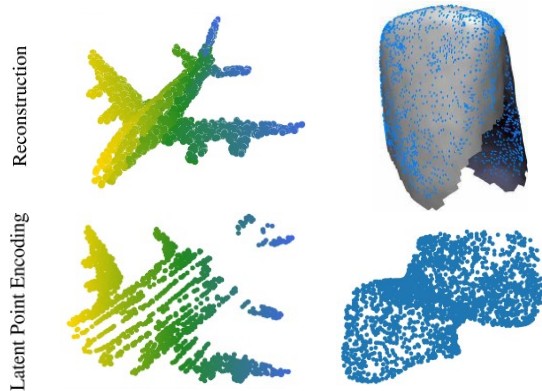

Figure 9: *Left*: While accurately reconstructed, the airplane forms a non-continuous distribution in the latent point encoding, posing challenges for sampling. *Right:* An incisor and its corresponding point encodings.

Fig. 9. The latent point encodings form a non-continuous distribution, posing challenges for sampling new models. Note that for point clouds sharing topology, VF-Net is strongly biased towards generating a continuous distribution; see Fig. 9. Addressing this issue could potentially be solved by training a flow or diffusion

prior for the point encodings, similar to the approach used in LION (Zeng et al., 2022). However, since this was not the focus of our model, we did not pursue this idea.

## 6 Conclusion

We have introduced the *FDI 16* dataset and *Variational FoldingNet (VF-Net)*, a fully probabilistic point cloud model in the same spirit as the original variational autoencoder (Kingma & Welling, 2014; Rezende et al., 2014). The key technical innovation is the introduction of a point-wise encoder network that replaces the commonly used Chamfer distance, allowing for probabilistic modeling. Importantly, we have shown that VF-Net offers better auto-encoding than current state-of-the-art generative models and more realistic sample generation for dental point clouds. Additionally, VF-Net offers straightforward shape completion and extrapolation due to its latent point encodings. All while identifying highly interpretable latent representations.

**Impact statement.** This paper contributes a generative model that is particularly suitable for dental data. This translates into several positive use cases within clinical practice. However, previous generative models have shown to be useful for less positive use cases such as deep fakes and fake news. It is unclear how this could take form in digital dentistry, but destructive minds tend to be creative.

**Acknowledgements.** This work was funded in part by the Novo Nordisk Foundation through the Center for Basic Machine Learning Research in Life Science (NNF20OC0062606). SH was supported in part by research grants (15334, 42062) from VILLUM FONDEN. JY was supports by Innovation Fund Denmark (1044-00172B).

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
