## S1    Supplementary Materials

### S1.1    Chamfer vs Euclidean

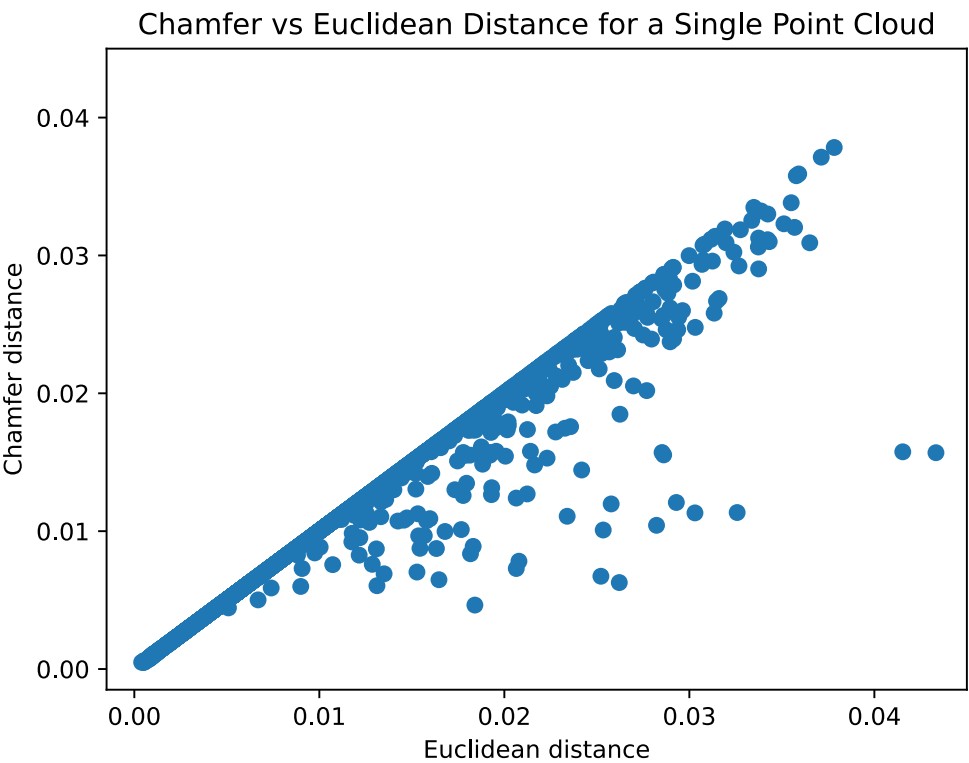

Figure S1: In the depicted plot, it is evident that the Euclidean distance serves as an upper limit for the Chamfer distance. In most instances, optimization with VF-Net results in nearly identical distances. This empirical finding supports our claim that the Chamfer distance can be efficiently replaced with an appropriate encoder choice.

### S1.2    ShapeNet Reconstruction Performances

Table S1: Both Chamfer distances (CD) and earth mover's distances (EMD) are multiplied by 1000. Lower values indicate better reconstruction performance.

| Method | ShapeNet Airplanes | |
| --- | --- | --- |
| | **CD** | **EMD** |
| DPM | 0.18 | 47.82 |
| SetVAE | 0.14 | 30.60 |
| PVD | 0.31 | 90.45 |
| LION | **0.025** | **7.30** |
| FoldingNet | 0.079 | 31.47 |
| VF-Net (ours) | 0.039 | 7.90 |

### S1.3  Data Handling

For dental scan experiments, the point clouds used were constructed from vertices and facet midpoints. The cardinality of the raw point clouds varied significantly, ranging from around 2,000 to 65,000 points. To handle this, we subsampled 2048 points from each point cloud during training. As FoldingNet and VF-Net deform from a set space, selecting an appropriate normalization method is crucial. The normalization determines the amount of deformation needed for initial points to reach the desired final reconstruction. As we have defined our planar patch as $[-1, 1]^2$, we scale the data so 99.5%

### S1.4  FDI 16 Training Details

VF-Net was trained using an Adamax optimizer with an initial learning rate of 0.001. Each backpropagation iteration utilized a batch size of 64, and the training persisted for 16,000 epochs. We employed a KLD warm-up (Sønderby et al., 2016) during the initial 4,000 epochs, whenever applicable. During the initial training of VF-Net, a constant variance was used. Following that, a distinct training phase of 100 epochs was explicitly conducted to train the variance network.(Detlefsen et al., 2019).

### S1.5  All FDI Training Details

In the experiments conducted on the proprietary dataset, encompassing all teeth, each model maintained the same architecture and size as employed in the FDI 16 experiment. However, training was confined to 1,250 epochs, incorporating a KLD warm-up phase constituting one-fourth of the total epochs when relevant. Again, a separate training run of 100 epochs was done to tune the variance network (Detlefsen et al., 2019).

### S1.6  Sampled All FDI Teeth

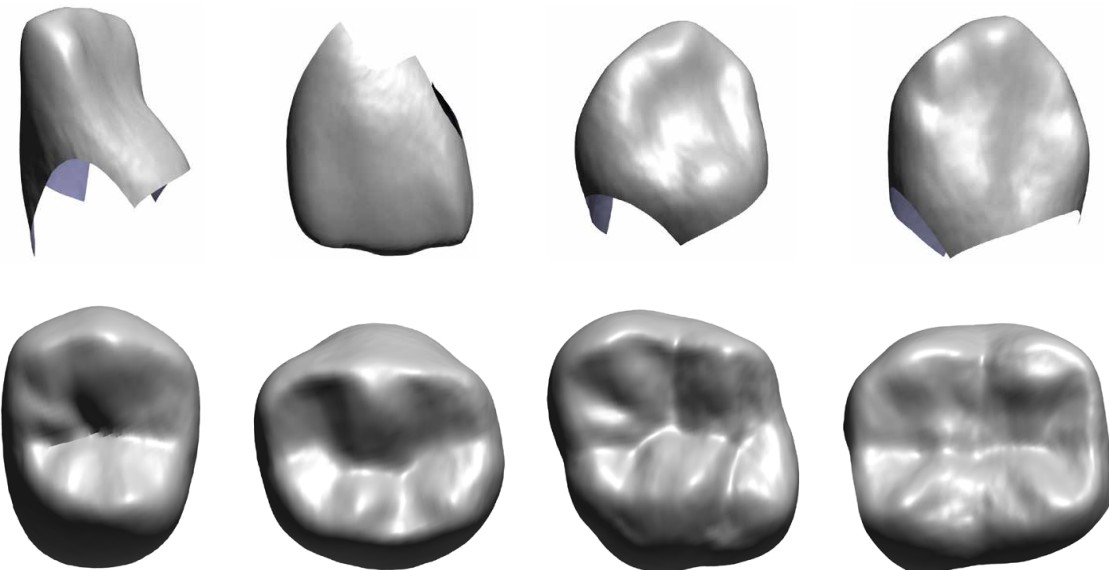

Figure S2: A display of meshes sampled by VF-Net showcases comprehensive coverage across the four major tooth modalities: Incisor, canine, pre-molar, and molar. These represent the primary types of teeth in the proprietary dataset.

### S1.7 Shape Completion Experiments

In the shape completion experiments, each model is permitted to sample three times the standard number of points. Unsupervised models, therefore, have the allowance to sample 6,144 points each, as their sampling is not limited to the missing area. Conversely, for supervised models trained to predict the 200 missing points, we extract 600 points in this region to balance point density between supervised and unsupervised models. The evaluation is done by calculating a one-directional Chamfer distance (1) from predicted points to ground truth, quantifying shape completion performance.

### S1.8 Interpolation

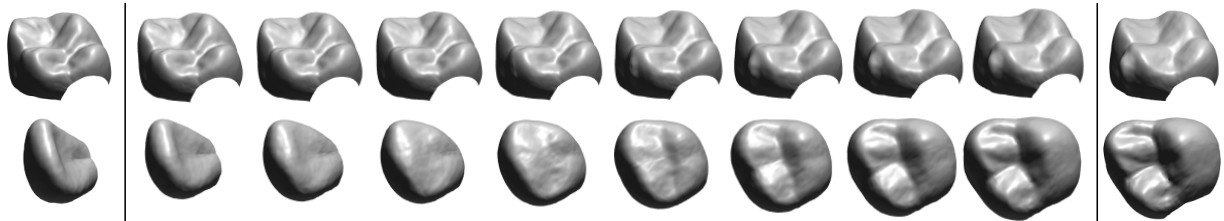

Figure S3: Interpolating between two teeth by interpolating their latent codes using the same mesh decoding.

### S1.9 Synthetic Toothwear Teeth

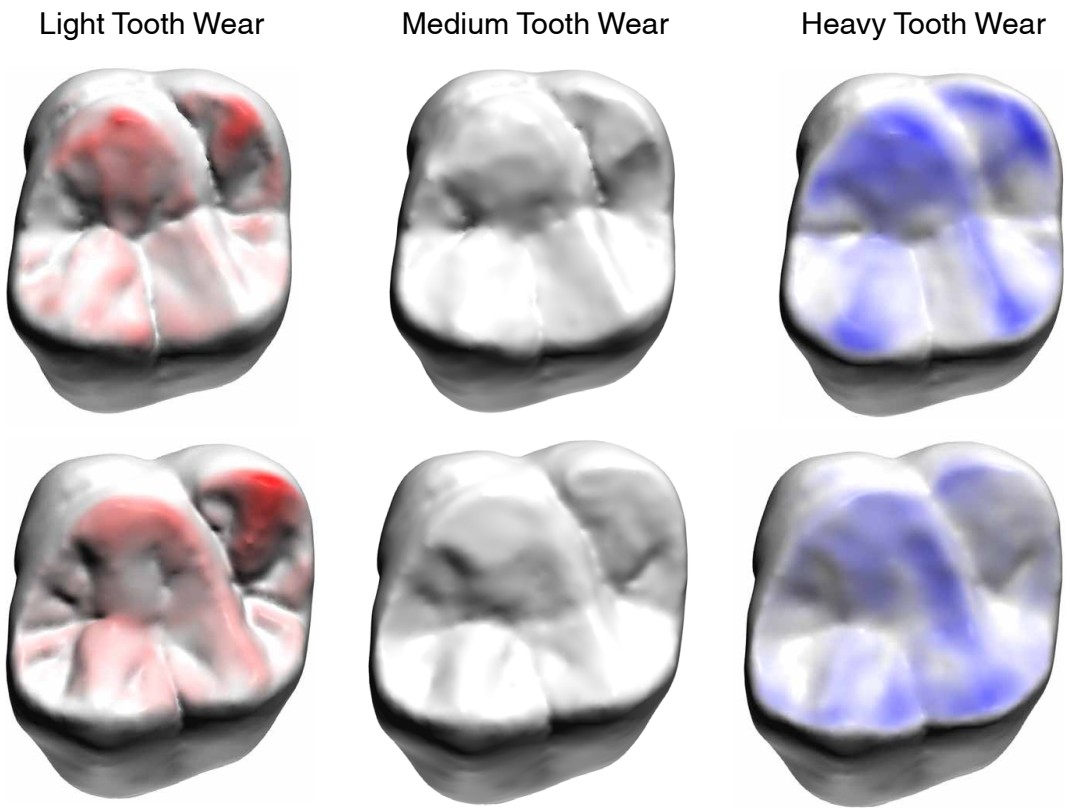

Figure S4: Two of the ten manually sculpted teeth to simulate tooth wear. *Left*: Areas with higher values compared to the original mesh are highlighted in red. *Middle*: The original mesh. *Right*: The mesh showcases areas depicted in blue that are lower than the original mesh.