# OpenReview forum: "Variational Autoencoding of Dental Point Clouds"
_TMLR — Accepted by TMLR_

### Review · Reviewer_efj1 · 2024-06-17

**Summary Of Contributions:**

This work proposes a VAE-based generative model for dental point clouds. A single point cloud here corresponds to a single tooth. Its contributions are threefold:
- The authors propose VF-Net, a variational autoencoder for point clouds. To my understanding (which is incomplete - see weaknesses), the unique idea here is to build a VAE using an autoencoding structure inspired by FoldingNet, which contains two types of latent variables: $\mathbf{z}$, which captures shape structure, and $\mathbf{g}$, which capture individual points' locations on the shape.
- The authors also introduce a new dataset, FDI 16, of 7732 tooth meshes.
- This appears to be the first application of generative shape modelling to digital dentistry. The proposed model is state-of-the-art at modelling teeth.

**Audience:**

Yes

**Claims And Evidence:**

No

**Requested Changes:**

## Critical changes
- Please take some time to clean up the notation, especially in Section 3, to make it more understandable. Take any approach you find to be the most clear, but in my opinion, a bulletproof way to remove ambiguity would be to include all of the following:
  - All the key variables (eg. $\mathbf{x}$, $\mathbf{g}$, $\mathbf{z}$, $\mathcal{G}(\mathbf{x},\mathbf{z})$), clearly defined, ideally including the set to which they belong.
  - All the functions introduced in the section, including the encoder, decoder, and "projection" function, with clearly defined domains and codomains.
  - The form of the approximate posterior $q(\mathbf{z} | \mathbf{x})$ (along with $p(\mathbf{x} | \mathbf{z})$ and $p(\mathbf{z})$, which are already included)
   - The loss used to train the model, expressed using the functions and key variables you've defined as per the previous three bullets.

## Minor but necessary changes
- Figure 1 caption: anatomically -> anatomical
- The assertion that Chamfer Distances cannot be used in probabilistic modelling is overly strong - "probabilistic modelling" can be interpreted quite broadly and not every distribution used in "probabilistic modelling" needs to (1) be Euclidean-valued or (2) have a tractable normalizing constant.
- Figure 2 caption: "Figure 2" appears twice
- Figure 2 mentions that the encoder is "graph-based," but the main text never mentions this anywhere.
- Page 2: planer -> planar
- Equation 2, RHS: this should read CHAMF-DIST$(\mathbf{x}, f((e(\mathbf{x}), \mathcal{G})))$ , not CHAMF-DIST$(\mathbf{x} - f((e(\mathbf{x}), \mathcal{G})))$
- The meaning of the notation $\mathbf{x}$ seems ambiguous. In equation (1) it represents an individual point in a point cloud, whereas in equation (2) it appears to represent an entire point cloud. Later it seems to represent an individual point again, though equation (5) is again ambiguous.
- Weird line spacing on page 8 in the "Simulated Shape Completion" paragraph

## Non-critical changes
- The work doesn't contain any comparisons to previous algorithms designed specifically for digital dentistry. It would be really cool if VF-Net were to outperform more bespoke tooth modelling algorithms, such as the work of Qiu et al. (2013), which I think is applicable to the "Simulated Shape Completion" paragraph in Section 5. It would be helpful, then, to see Qiu et al. (2013) added to Table 4.
- "Probabilistic evaluation" is a nonstandard term - either more precise terminology or some other added explanation would be helpful.
- One of the model's limitations is that generated teeth are overly smooth. It would be nice to see some discussion about the downstream effects of this limitation in actual applications; I assume it would make the model less likely to capture the distribution of teeth with highly localized defects, for example.

**Strengths And Weaknesses:**

## Weaknesses
1. The main weakness in my view is that the method description (Section 3) is difficult to understand.
     - For one, some of the notation is all over the place. For example, $\mathcal{G}$ is a latent space, $\hat{\mathcal{G}}$ is a finite set of latent points, and $\hat{\mathcal{G}}(\mathbf{x}, \mathbf{z})$ is a specific "projected" point in the set $\hat{\mathcal{G}}$. These are confusing for a couple reasons.
       - Referring to specific points as $\hat{\mathcal{G}}(\mathbf{x}, \mathbf{z})$ is misleading in light of the fact that a lowercase $\mathbf{g}$ is otherwise used for generic points in the same set.
       - The $\hat{\mathcal{G}}(\mathbf{x}, \mathbf{z})$ notation for a projection may or may not mean the same thing as a different notation introduced beforehand: $\text{proj}_\mathcal{G}(\mathbf{x}): \mathbb{R}^3 \to \mathcal{G}$.

       These are just a couple of examples of where notational choices could be more consistent and suggestive of the meaning they signify. The fact that most of these notations are introduced in one paragraph and then never used again makes it even harder to infer what's going on.
     - How are latent point encodings $\mathbf{g}$ obtained? There seem to be two ways to do this:
       1. Through the encoder (Figure 2 seems to suggest that point encodings are obtained through "local feature vectors").
       2. Through the projection $\text{proj}_\mathcal{G}(\mathbf{x})$ (or maybe $\hat{\mathcal{G}}(\mathbf{x}, \mathbf{z})$) given by Equation (4).
     - No loss is given for this model. Equation (5) outlines a generic ELBO loss, but that doesn't apply to this model because, for example, $p(\mathbf{x} | \mathbf{z})$ is never defined. Only $p(\mathbf{x} | \mathbf{z}, \mathbf{g})$ is defined. Is $p(\mathbf{x} | \mathbf{z}) := p(\mathbf{x} | \mathbf{z}, \text{proj}_\mathcal{G}(\mathbf{x}))$?
     - The structure of the encoder is discussed, but not how it is used to parameterized the approximate posterior $q(\mathbf{z} | \mathbf{x})$.
     - Most of the discussion and notation in Section 3 pertains to individual points $\mathbf{x}$, not entire point clouds $\mathbf{X}$. The section never makes clear how the the full point cloud structure $\mathbf{X}$ is incorporated into the optimization process.

2. A key advantage boosted by the paper is that VF-Net computes the likelihood using the L2 loss rather than the Chamfer distance, which lacks a tractable normalizing constant but is used by competing methods. I question how much of an advantage this really is; it only really affects the coefficient of the reconstruction term in the loss, right? The authors put forward "probabilistic evaluation" as an advantage; I assume this means the ability to evaluate true densities/likelihoods/ELBOs. Does that actually have utility in this setting?

3. As the authors point out, the model generates excessively smooth shapes (akin to VAEs in other domains).

## Strengths
- Outside of Section 3, the paper is clear and well-motivated. I like that the authors explain clearly how shape modelling is useful in computational dentistry: tooth generation, shape completion, diagnostics, etc.
- By multiple metrics, VF-Net generates more realistic teeth than other shape generation algorithms.
- VF-Net also demonstrates some interesting domain-specific capabilities, like point-wise uncertainty quantification and simulated shape completion.
- The representation learning experiments were also cleverly designed, in my opinion. They clearly show that the learned representations are more robust than FoldingNet.
- It's always nice to see a new dataset released alongside a paper. Hopefully this will open up future research in digital dentistry.

---

> ### Author Response · Authors · 2024-06-25
>
> >Critical Changes
>
> We thank the reviewer for their detailed review and apologize for any confusion caused by the initial manuscript. In response to the reviewer's suggestions, we have significantly rewritten Section 3, particularly Section 3.1, to ensure clarity and precision. Outside clarifying the mathematical notation, we have made efforts to clarify the relationship between the encoder, the projections, and the latent code. We hope that these revisions address the reviewer's concerns and make the new section more comprehensible. Please let us know if further clarifications or modifications are needed.
>
> >Figure 1 caption: anatomically -> anatomical
>
> Thank you very much for your thorough review and for pointing out the typos in our manuscript. We have corrected it accordingly.
>
> >The assertion that Chamfer Distances cannot be used in probabilistic modelling is overly strong - "probabilistic modelling" can be interpreted quite broadly and not every distribution used in "probabilistic modelling" needs to (1) be Euclidean-valued or (2) have a tractable normalizing constant.
>
> We appreciate the reviewer’s feedback regarding the use of Chamfer Distances in probabilistic modeling. We agree that probabilistic modeling is a broad term and can encompass various approaches, not all of which require Euclidean values or tractable normalizing constants.
>
> However, in the context of Variational Autoencoders (VAEs), explicit likelihood evaluation with a tractable normalization constant is a key characteristic. This is essential for the VAE framework to function correctly, as it relies on maximizing the Evidence Lower Bound (ELBO) of the marginal likelihood. The use of Chamfer Distance does not provide a straightforward way to compute a tractable likelihood, which is why we distinguish our approach as being fully probabilistic in the traditional VAE sense.
>
> >Figure 2 caption: "Figure 2" appears twice
>
> We have corrected it so “Figure 2” only appears once in the caption
>
> >Figure 2 mentions that the encoder is "graph-based," but the main text never mentions this anywhere.
>
> The term "graph-based encoder" is derived from the FoldingNet literature. To maintain consistency and clarity, we have updated the terminology in both Figure 2 and the main text to refer to the encoder as a "PointNet encoder."
>
> >Page 2: planer -> planar
>
> We have corrected it to “planar”
>
> >Equation 2, RHS: this should read CHAMF-DIST$(\mathbf{x}, f((e(\mathbf{x}), \mathcal{G})))$ , not CHAMF-DIST$(\mathbf{x} - f((e(\mathbf{x}), \mathcal{G})))$
>
> We thank the reviewer for their attention to detail. We have corrected it so the notation is consistent.
>
> >The meaning of the notation $\mathbf{x}$ seems ambiguous. In equation (1) it represents an individual point in a point cloud, whereas in equation (2) it appears to represent an entire point cloud. Later it seems to represent an individual point again, though equation (5) is again ambiguous.
>
> We appreciate the reviewer's attention to detail and feedback. We apologize for any confusion caused by the ambiguity in the notation. Accordingly, we have thoroughly revised section 3 and the mentioned equations to hopefully clear up any confusion. Please let us know if further details are required.
>
> >The meaning of the notation $\mathbf{x}$ seems ambiguous. In equation (1) it represents an individual point in a point cloud, whereas in equation (2) it appears to represent an entire point cloud. Later it seems to represent an individual point again, though equation (5) is again ambiguous.
>
> We thank the reviewer for their attention to detail. We apologize for any confusion this notation may have caused. We have accordingly changed the notation in equation 1 to be more consistent with the remaining manuscript.
>
> >Weird line spacing on page 8 in the "Simulated Shape Completion" paragraph
>
> We have corrected this.

---

> ### Author Response · Authors · 2024-06-25
>
> >The work doesn't contain any comparisons to previous algorithms designed specifically for digital dentistry. It would be really cool if VF-Net were to outperform more bespoke tooth modelling algorithms, such as the work of Qiu et al. (2013), which I think is applicable to the "Simulated Shape Completion" paragraph in Section 5. It would be helpful, then, to see Qiu et al. (2013) added to Table 4.
>
> We appreciate the reviewer's suggestion to include comparisons with previous algorithms designed specifically for digital dentistry, such as the work of Qiu et al. (2013). However, we were unable to include a direct comparison with the method proposed by Qiu et al. (2013) because no public implementation of their algorithm is available.
>
> >One of the model's limitations is that generated teeth are overly smooth. It would be nice to see some discussion about the downstream effects of this limitation in actual applications; I assume it would make the model less likely to capture the distribution of teeth with highly localized defects, for example.
>
> We have added a discussion on this issue at the beginning of the limitations section:
>
> "Limitations: Similar to variational autoencoders in other domains, VF-Net tends to produce overly smooth samples. This characteristic could impact applications such as crown generation, where precise replication of the biting surface is crucial to prevent patient discomfort. Moreover, the model's tendency towards smoothness suggests potential challenges in capturing finer details of teeth, which are essential for comprehensive representation learning."
>
> These insights highlight areas where future improvements in model architecture or training strategies could enhance the fidelity of generated dental structures."

---

> > ### Comment · Reviewer_efj1 · 2024-06-28
> >
> > Thank you for comprehensively addressing my concerns and the thorough rewrite of the method section, which has made it much clearer. That addresses most of my reservations with the exception of some lingering notational ambiguity.
> >
> > - Based on my reading of Equation (1), the newest adjustments use $\mathbf{x}$ for a point cloud and $x$ or $x_i$ for individual points. Thanks for making this clearer, but this notation is still not followed in Equation (6).
> >
> > - Also, there's some contradiction in the way $\mathbf{g}$ is defined; based on wording in 3.1, I believe $\mathbf{g}$ is a collection of projected points corresponding to the point cloud $\mathbf{x}$, but equation (3) implies that $\mathbf{g} \in \mathcal{G} = [0, 1]^2$, meaning $\mathbf{g}$ must be a single point. Which is true? If $\mathbf{g}$ were a single point in $\mathcal{G}$, why is it being computed from an entire point cloud $\mathbf{x}$?
> >
> >     Even if I assume this is a typo and $\mathbf{g}$ is meant to be a collection of points (each of which is in $\mathcal{G}$), there's still some ambiguity in Equation (3). I believe Equation (3) is meant to be "pointwise" in that it refers to the way specific points $g_i$ and $x_i$ rather than collections $\mathbf{g}$ and $\mathbf{x}$. This should be made explicit.
> >
> > - As a general comment, Section 3.1 would benefit from more explicit definitions of the variables involved. As it stands we need to infer the meaning of notation from the equations in the paper. It would be easy enough to add "Assume $\mathbf{x}$ is a point cloud of points $x_i$ ..., let $\mathbf{g}$ be the set of corresponding point encodings $g_i$.." etc. .
> >
> > - The distribution of the approximate posterior $q(z | x)$ is not defined - is it Gaussian?

---

> ### Author Response · Authors · 2024-06-29
>
> We would like to thank the reviewer spending their time engaging in discussion. We find their feedback valuable and helpful to improving our manuscript. We have uploaded a new version of the manuscript with revisions
>
> >Based on my reading of Equation (1), the newest adjustments use for a point cloud and or for individual points. Thanks for making this clearer, but this notation is still not followed in Equation (6).
>
> We appreciate the reviewer's attention to detail and their valuable feedback. We apologize for the oversight in our notation. We have now revised Equation (6) to ensure it follows the same notation used in Equation (1), providing consistency throughout the manuscript.
>
> >Also, there's some contradiction in the way is defined; based on wording in 3.1, I believe is a collection of projected points corresponding to the point cloud , but equation (3) implies that , meaning must be a single point. Which is true? If were a single point in , why is it being computed from an entire point cloud?
> Even if I assume this is a typo and is meant to be a collection of points (each of which is in ), there's still some ambiguity in Equation (3). I believe Equation (3) is meant to be "pointwise" in that it refers to the way specific points and rather than collections and . This should be made explicit.
>
> We thank the reviewer for their insightful feedback and for highlighting the ambiguity in the notation. We apologize for any confusion caused by the inconsistency. Based on the reviewer's comments, we have revised Equation (3) to explicitly denote it as point-wise, ensuring clarity and consistency with the description in Section 3.1.
>
> >As a general comment, Section 3.1 would benefit from more explicit definitions of the variables involved. As it stands we need to infer the meaning of notation from the equations in the paper. It would be easy enough to add "Assume is a point cloud of points ..., let be the set of corresponding point encodings .." etc.
>
> We appreciate the reviewers suggestions and have added a sentence introducing the variables to the manuscript accordingly
>
> >The distribution of the approximate posterior is not defined - is it Gaussian?
>
> Yes, the approximate posterior still follows a gaussian distribution. We have accordingly added explicit mention of this to the manuscript.

---

> > ### Comment · Reviewer_efj1 · 2024-07-04
> >
> > Thanks for responding to my feedback. All my concerns have been addressed now.

---

### Review · Reviewer_HU4R · 2024-06-19

**Summary Of Contributions:**

The contributions of this paper are summarised as follows:

1. The paper proposes a new VAE method for representation learning and generation of 3D point clouds, with a focus on a dental point cloud dataset. The proposed method appears to achieve better performance than a few recent methods on the dataset.

2. The paper introduces a new 3D dental point cloud dataset.

**Audience:**

Yes

**Broader Impact Concerns:**

Broader Impact Statement has been presented in the paper.

**Claims And Evidence:**

Yes

**Requested Changes:**

1. More discussions and/or comparions to the methods in [1, 2] and adjustment to the claims of the contributions accordingly.

2. Improvement to clarity.

**Strengths And Weaknesses:**

Strengths:

1. The experiments of the paper look relatively comprehensive and several evaluations are covered, including point cloud generation, shape completion, representation learning, etc. The baselines seem to be recent advances in the area.

2. The release of a new dataset is highly welcome, which can be valuable to the research area.

Weaknesses:

1. The focus of the paper is on the newly introduced dental point cloud dataset. It is a bit unclear about the unique characteristics of the new dataset compared with existing point cloud datasets. For example, does the new dataset bring new research challenges that existing method cannot address and can be addressed by the proposed method? If not, it might be interesting to evaluate the proposed method on other benchmark datasets as well.

2. The paper claims that the proposed method is the first fully probabilistic VAE for point clouds. However, there are existing VAE methods such as in [1, 2]. Given the existing works, the claim might be bold to me. Moreover, the discussions and comparions to the following methods are also needed.

[1] Anvekar, Tejas, et al. "VG-VAE: a venatus geometry point-cloud variational auto-encoder." Proceedings of the IEEE/CVF conference on computer vision and pattern recognition. 2022.

[2] Li, Shidi, Miaomiao Liu, and Christian Walder. "EditVAE: Unsupervised parts-aware controllable 3D point cloud shape generation." Proceedings of the AAAI Conference on Artificial Intelligence. Vol. 36. No. 2. 2022.

3. The clarity of the paper needs to be improved significantly.
- A few details are missing in the paper, including: what's the architectures of the encoder and decoder (I guess the paper reuses those from FoldingNet, but it is important to introduce the details)? What's the prior and variational posterior of $z$?
- A few configurations of the method are not justified or motivated or supported with ablation study, including why the projection in Eq 3 and 4 is useful? Why student-t distribution is used as the likelihood function?
- It is hard to understand the core part of the proposed method. Specifically, Eq 4 is a minimisation. How is this minimisation incorporated in ELBO of VAE? Eq 4 is also a reconstruction term of the input $x$. Why is a square error used here?

---

> ### Author Response · Authors · 2024-06-25
>
> >The focus of the paper is on the newly introduced dental point cloud dataset. It is a bit unclear about the unique characteristics of the new dataset compared with existing point cloud datasets. For example, does the new dataset bring new research challenges that existing method cannot address and can be addressed by the proposed method? If not, it might be interesting to evaluate the proposed method on other benchmark datasets as well.
>
> We appreciate the reviewer's query regarding the unique characteristics of the FDI 16 dataset. The FDI 16 dataset indeed presents a different challenge compared to existing point cloud datasets:
> * Real-World Data: Unlike idealized CAD scans commonly used in research (e.g. in the ShapeNet and ModelNet datasets), the FDI 16 dataset is based on real-world dental scans. This introduces realistic variances and imperfections that are not present in synthetic datasets.
> * Intermediate Complexity: The dataset is less complex and sparse than lidar scans, providing an excellent middle-ground for testing methods.
> * Partial Occlusions: A critical feature of the dataset is the presence of partial occlusions, particularly on the mesial and distal sides of the teeth (the sides commonly with neighboring teeth). This requires robust handling of incomplete data, a challenge our method addresses effectively.
> * Irregular Sampling: The dataset is irregularly sampled, with higher point density in areas of greater curvature. This characteristic necessitates advanced techniques for handling non-uniform point distributions.
> * Sparse Meshes: The FDI 16 dataset meshes are much sparser than those from CAD scans, resulting in lower connectivity and posing a challenge for methods reliant on dense, highly connected meshes.
>
> These unique characteristics create specific research challenges that our proposed method is designed to address. We believe these features differentiate the FDI 16 dataset from existing datasets and highlight the importance of evaluating our method on this data.
>
> >The paper claims that the proposed method is the first fully probabilistic VAE for point clouds. However, there are existing VAE methods such as in [1, 2]. Given the existing works, the claim might be bold to me. Moreover, the discussions and comparions to the following methods are also needed.
>
> We value the reviewer's feedback. While it is true that the methods mentioned in [1, 2] utilize VAE-like frameworks, they do not explicitly minimize a likelihood, which we consider a defining characteristic of a VAE. Instead, these methods approximate the reconstruction term using the Chamfer distance. In contrast, our proposed method explicitly incorporates likelihood maximization within the VAE framework.
>
> We acknowledge the need for a more detailed discussion and comparison with these existing methods in the revised manuscript under the related works section. This will provide a clearer context for our claim of being the first fully probabilistic VAE for point clouds.
>
> >A few details are missing in the paper, including: what's the architectures of the encoder and decoder (I guess the paper reuses those from FoldingNet, but it is important to introduce the details)? What's the prior and variational posterior of $z$?
>
> We appreciate the reviewer's feedback and their valuable suggestions for clarification. We have now included explicit details regarding the architecture of both the encoder and decoder in the revised manuscript in Section 3.1.
> Regarding the prior and variational posterior, we use a normalizing flow as the prior distribution. The variational posterior is derived using FoldingNet's encoder, which captures the input point cloud's features and maps them to the latent space z.

---

> ### Author Response · Authors · 2024-06-25
>
> >A few configurations of the method are not justified or motivated or supported with ablation study, including why the projection in Eq 3 and 4 is useful? Why student-t distribution is used as the likelihood function?
>
> Regarding the projection in Equations 3 and 4, it is essential to maintain a one-to-one mapping throughout our network. Without this, we would not be able to evaluate in a student-t distribution as our likelihood function. We have significantly revised Section 3 in the manuscript to better justify and motivate our design decisions.
>
> As for the choice of the Student-t distribution as our likelihood function, we opted for this distribution to better handle the characteristics of point cloud data. Specifically, the Student-t distribution allows the model to focus more on the central mode of the points while being robust to outliers. This decision is supported by insights from related studies, such as those discussed in [3], which highlight the effectiveness of the Student-t distribution in modeling data distributions with heavy tails.
>
> [3] Hiroshi Takahashi, Tomoharu Iwata, Yuki Yamanaka, Masanori Yamada, and Satoshi Yagi. Student-t Variational Autoencoder for Robust Density Estimation. In Proceedings of the Twenty-Seventh International Joint Conference on Artificial Intelligence, pp. 2696–2702, Stockholm, Sweden, July 2018. International Joint Conferences on Artificial Intelligence Organization. ISBN 978-0-9992411-2-7. doi: 10.24963/ijcai.2018/374. URL https://www.ijcai.org/proceedings/2018/374.
>
> > It is hard to understand the core part of the proposed method. Specifically, Eq 4 is a minimisation. How is this minimisation incorporated in ELBO of VAE? Eq 4 is also a reconstruction term of the input $x$. Why is a square error used here?
>
> We appreciate the reviewer’s questions regarding Equation 4 and its role in our proposed method. To clarify:
>
> Equation 4 represents the optimal projection for the input x and is minimized implicitly as part of the likelihood term in the ELBO framework. This allows us to use the classical evidence lower bound framework. In our approach, Equation 4 serves as a standard notation for the reconstruction error, commonly expressed as a square error in many variational autoencoder (VAE) frameworks. Note that this does not affect the optimization process since it is not explicitly evaluated or minimized.
>
> Please note that Section 3 has been extensively rewritten with added explanations regarding the context and implementation of Equation 4. We believe these revisions provide a clearer understanding of its role in our methodology. Should the reviewer require further clarification, we are prepared to provide additional details.

---

### Review · Reviewer_b4Xv · 2024-06-19

**Summary Of Contributions:**

The paper introduces Variational FoldingNet (VF-Net), a novel model that improves point cloud processing by utilizing a probabilistic encoder instead of traditional Chamfer distance minimization, enhancing computational efficiency and simplifying probabilistic modeling. It demonstrates state-of-the-art performance in dental reconstruction and interpolation tasks, achieving lower reconstruction errors. Additionally, the paper contributes to the field by releasing the FDI 16 Tooth Dataset, a comprehensive resource for future research in digital dentistry.

**Audience:**

Yes

**Broader Impact Concerns:**

The authors already included an Impact Statement.

**Claims And Evidence:**

Yes

**Requested Changes:**

My main requests regard the discussion of the reproducibility of some of the experiments and a more thorough discussion of the applicability of Diffusion Models in the field.

**Strengths And Weaknesses:**

## Strengths
- **Dataset Contribution:** The release of the FDI 16 Tooth Dataset is a valuable contribution to the field. This dataset provides a rich resource for future research in digital dentistry.
- **Approach:** The introduction of Variational FoldingNet (VF-Net) effectively replaces the traditional Chamfer distance minimization with a probabilistic encoder, increasing computational efficiency and simplifying the probabilistic modeling.
- **Empirical Performance:** The paper demonstrates lower reconstruction errors and state-of-the-art performance in dental sample generation, showcasing the superiority of VF-Net in dental reconstruction and interpolation tasks.

## Weaknesses
- **Reproducibility:** The dataset release is a valuable contribution; however, the use of a private dataset for certain experiments, particularly in the "Representation Learning" section, limits the study's reproducibility. Additionally, while annotations are provided for the proprietary dataset, they appear to be missing for the openly released dataset, which further hinders replication efforts.
- **Discussion of diffusion models:** The authors discuss the non-compression of the space of diffusion models; however, they overlook approaches such as Latent Diffusion Model [1], which effectively addresses this issue. Including a discussion on these methods would enhance the study.

[1] Rombach, Robin, et al. "High-resolution image synthesis with latent diffusion models. 2022 IEEE." CVF Conference on Computer Vision and Pattern Recognition (CVPR). 2021.

---

> ### Author Response · Authors · 2024-06-25
>
> >Reproducibility: The dataset release is a valuable contribution; however, the use of a private dataset for certain experiments, particularly in the "Representation Learning" section, limits the study's reproducibility. Additionally, while annotations are provided for the proprietary dataset, they appear to be missing for the openly released dataset, which further hinders replication efforts.
>
> We appreciate the reviewer’s feedback regarding the reproducibility of our study. We agree that the use of proprietary datasets for a few experiments is problematic. We, however, consider the obtained results to be sufficiently interesting to justify their inclusion.
>
> >Discussion of diffusion models: The authors discuss the non-compression of the space of diffusion models; however, they overlook approaches such as Latent Diffusion Model [1], which effectively addresses this issue. Including a discussion on these methods would enhance the study.
>
> We appreciate the reviewer’s comment and apologize for the lack of clarity on our part. LION [2] and FrePolad [3] are two latent diffusion models, which we discussed in our initial submission. To clarify this point, we have moved them from the “Existing Point Cloud Variational Autoencoders” section to the “Other Generative Models” section under related work, where we also explicitly call them latent diffusion models..
>
>
> [2] Xiaohui Zeng, Arash Vahdat, Francis Williams, Zan Gojcic, Or Litany, Sanja Fidler, and Karsten Kreis. LION: Latent Point Diffusion Models for 3D Shape Generation, October 2022. URL http://arxiv.org/abs/2210.06978. arXiv:2210.06978 [cs, stat].
>
> [3] Chenliang Zhou, Fangcheng Zhong, Param Hanji, Zhilin Guo, Kyle Fogarty, Alejandro Sztrajman, Hongyun Gao, and Cengiz Oztireli. FrePolad: Frequency-Rectified Point Latent Diffusion for Point Cloud Generation, November 2023. URL http://arxiv.org/abs/2311.12090. arXiv:2311.12090 [cs].

---

### Author Response · Authors · 2024-06-25

Dear Reviewers,

Thank you for your valuable feedback and insightful comments. We have found them very helpful. After carefully revising the manuscript to address your concerns and suggestions, we have re-uploaded the new version. We look forward to further discussion.

Best regards,

Submission 2770 Authors

---

### Decision · Action_Editor_xfua · 2024-07-18

**Recommendation:** Accept as is

**Comment:**

The submission introduces a new dataset for digital dentistry and a novel method, Variational FoldingNet (VF-Net), as a probabilistic autoencoder for point clouds.  Reviewers originally had some concerns on the novelty of the method and the presentation quality. Post revision and discussion, all reviewers agreed that the revised paper has addressed their concerns and recommended acceptance. The AE agreed with the recommendation.

**Audience:**

Yes, the AI for health community may be interested.

**Claims And Evidence:**

The paper introduces a new dataset for digital dentistry and a novel method, Variational FoldingNet (VF-Net), as a probabilistic autoencoder for point clouds.  Reviewers agreed that the evidence supports these claims---the usefulness of the dataset and the novelty of the new method.

---

> ### Author Response · Authors · 2024-08-02
>
> We thank the reviewers and editor for their valuable feedback throughout the review process. We have uploaded a camera-ready version of our submission accordingly.